# Seroprevalence of *Getah virus* in Pigs in Eastern China Determined with a Recombinant E2 Protein-Based Indirect ELISA

**DOI:** 10.3390/v14102173

**Published:** 2022-09-30

**Authors:** Qing Sun, Yixuan Xie, Zhixin Guan, Yan Zhang, Yuhao Li, Yang Yang, Junjie Zhang, Zongjie Li, Yafeng Qiu, Beibei Li, Ke Liu, Donghua Shao, Jiaxiang Wang, Zhiyong Ma, Jianchao Wei, Peng Li

**Affiliations:** 1College of Animal Science, Yangtze University, Jingzhou 434025, China; 2Shanghai Veterinary Research Institute, Chinese Academy of Agricultural Sciences, Shanghai 200241, China

**Keywords:** serological survey, *Getah virus*, ELISA, China

## Abstract

*Getah virus* (GETV), in the genus *Alphavirus* and the family *Togaviridae*, has been detected throughout the world. GETV causes high morbidity and mortality in newborn piglets, entailing serious economic losses. Therefore, the experimental work on GETV detection is necessary. However, due to the influence of a variety of unavoidable factors, the ELISA test for the primary screening of animal diseases has low accuracy in detection results. Therefore, we optimized a recombinant E2 (rE2) protein-based enzyme-linked immunosorbent assay (ELISA) for the detection of GETV antibodies in pig serum. The E2 protein was successfully expressed and purified with SDS-PAGE. A Western blotting analysis of sera from infected pigs showed strong reaction with a viral antigen of ~46 KDa corresponding to the E2 glycoproteins. By using chessboard titration and comparing the P/N values, we found that the optimal concentration of coated antigen was found to be 24.5 μg/mL, and the optimal dilution of serum specimens was 1:100. The best working dilution of the horseradish peroxidase (HRP)-conjugated goat anti-pig immunoglobulin (IgG) was 1:5000. The optimal coating conditions were 12 h at 4 °C. The optimal incubation conditions for serum specimens, blocking, and reaction with the secondary antibody were all 1 h at 37 °C. We also investigated the seroprevalence of GETV in 133 serum specimens collected in Eastern China, and 37.59% of the samples tested positive for anti-GETV IgG antibodies, indicating that the seroprevalence of GETV is high in pig populations in China. The seroprevalence was significantly lower in spring (April; 24.24%, 16/66) than in autumn (October; 50.75%, 34/67), which suggested that the presence of anti-GETV antibodies in pigs was seasonal. In conclusion, we improved an rE2 ELISA that detected pig antibodies against GETV after experimental and natural infections. This should be useful in the diagnosis and surveillance of GETV infections.

## 1. Introduction

*Getah virus* (GETV) causes one of the most severe worldwide infectious diseases in swine of all ages [1]. GETV was originally isolated from *Culex gelidus* mosquitoes in Malaysia in 1955, and has since been reported in East Eurasia and South-East Asian countries [2,3,4,5,6], including Far East Russia, Mongolia, China, South Korea, Japan, Thailand, the Philippines, and Australia [4,7]. GETV infects domestic animals through the bites of mosquitoes, causing symptomatic disease [8,9]. The major mosquito vector of GETV is known to be *C. tritaeniorhynchus* Giles, which is also a major vector of Japanese encephalitis virus (JEV; family *Flaviviridae*, genus *Flavivirus*) [10]. Infection with GETV, a member of the genus *Alphavirus*, has been related to diarrhea and death in piglets, reproductive failure, and abortion in sows [11,12]. As with JEV, pigs infected with GETV become viremic, suggesting that pigs are the main amplifying host of GETV in nature [12]. A serological survey detected anti-GETV antibodies in animals such as birds, cattle, goats, and humans [13]. GETV infection can cause high morbidity and mortality in newborn piglets, leading to serious economic losses. Therefore, a rapid and effective method to evaluate GETV infection in pigs is necessary for its prevention and control. The GETV genome is a single-positive-stranded RNA of 11–12 kb, two-thirds of which encodes nonstructural proteins (NSP1–4) involved in viral RNA transcription, replication, polyprotein cleavage, and RNA capping; the remainder encodes structural proteins (C, E3, E2, 6K, and E1) [14]. Among these proteins, E2 occurs on the surface of the viral particle. E2 is the immunodominant region of the GETV E protein and plays a crucial role in the early steps of infection [15,16]. It mediates viral binding to the cell membrane and the subsequent fusion of the virus and cell [17]. The E2 region is on the outer surface of the virion envelope and induces immune protection against GETV infection in its host. Therefore, the E2 protein has potential utility as a diagnostic antigen to develop a specific and sensitive serological test.

Among the various serological methods currently used to detect GETV antibodies, the enzyme-linked immunosorbent assay (ELISA) is simple, effective, rapid, and economic, making it suitable for clinical applications. Compared with other detection methods, such as the fluorescent focus assay or immunofluorescent assay (IFA), ELISA is less expensive and less time-consuming and requires no additional laboratory apparatus. Various anti-GETV antibodies have been used to establish an ELISA for its detection. A recombinant E2 protein was used as the antigen for an indirect ELISA for horse GETV [18,19].

In this study, we expressed the GETV E2 protein in a prokaryotic expression system and improved a new ELISA based on the recombinant E2 protein (rE2). Our method can be used for the detection of anti-GETV immunoglobulin (IgG) in infected or inoculated pigs.

## 2. Materials and Methods

### 2.1. Serum Samples

A total of 133 clinical pig serum samples were collected from finishing pigs (24–28 weeks old) in slaughterhouses or sows (more than 28 weeks old) on pig farms in six provinces (Shandong, Hebei, Zhejiang, Shanghai, Jiangsu, Guangdong) in Eastern China, in April (*n* = 66) and in October (*n* = 67) of 2018. A total of 20 sera samples (GETV-P1~P10 and GETV-N1~N10) from naturally infected pigs, which were tested by VN [13], IFA [20], qRT-PCR [21], were used to establish and optimize the ELISA protocol (Appendix A). Positive sera of Japanese encephalitis virus (JEV, JE-P1~P5), porcine reproductive and respiratory syndrome (PRRSV, PRRS-P1~P5), classical swine fever virus (CSFV, CSF-P1~P5), Pseudorabies virus (PRV, PR-p1~p5), and porcine epidemic diarrhea virus (PEDV, PED-P1~P5) verified by VN and IFA (Appendix A) were from experimentally infected pigs provided by the China Animal Health and Epidemiology Center (Shanghai Branch) and were used to optimize ELISA protocols.

### 2.2. Expression and Purification of rE2 Protein

We referred to the complete gene sequence of structural protein E2 of GETV (SH05-6) in GenBank (EU015066), whose length is 1278bp, and optimized and synthesized the gene sequence according to the codon of *E. coli*. The recombinant plasmid pCold I-E2 was constructed (and is maintained by the Shanghai Veterinary Research Institute, Shanghai, China). The recombinant vector was confirmed with SDS-PAGE and the expressed protein with a Western blotting assay based on GETV-positive serum (GETV-P1) and a His-tagged antibody. The expressed protein was purified on the Ni-column using the His-Bind Purification Kit (Bio-Rad, Hercules, CA, USA), according to the instructions of the manufacturer, and was confirmed with SDS-PAGE.

### 2.3. ELISA Development

An indirect ELISA was carried out and optimized with positive (GETV-P2~P4) and negative (GETV-N1~N3) control serum samples. The purified rE2 protein was diluted with 0.05 mol/L carbonate buffer (pH 9.6). Then, 100 µL of the diluted antigen was added to each well of a 96-well ELISA plate and incubated at 4 °C for 12 h. After the plate was washed three times with phosphate-buffered saline (PBS) containing 0.05% Tween 20 (PBST), it was blocked at 37 °C for 1 h with 200 µL of 5% bovine serum albumin (BSA) dissolved by PBST. After blocking, the plates were washed three times with PBST and air-dried. Serum samples (100 μL/well) diluted with PBST containing 5% BSA were added to the plates and incubated for 60 min at 37 °C. After the plate was washed three times with PBST, 100 µL of diluted horseradish peroxidase (HRP)-conjugated goat anti-swine IgG antibody (Sigma-Aldrich, Burlington, MA, USA) was added and incubated at 37 °C for 1 h. The wells were then washed three times with 100 µL of PBST. The peroxidase reaction was visualized with 3,3’,5,5’-tetramethylbenzidine (TMB) solution as the substrate (KPL, Gaithersburg, MD, USA). The reaction was terminated by the addition of 100 µL of 2 M sulfuric acid to each well and the optical density at a wavelength of 450 nm (OD_450_) of each well was read with a microplate reader (Thermolab System, Helsinki, Finland).

Under these basic conditions, the antigen coating concentration, the serum dilution ratio, the coating conditions, the blocking conditions, and the serum conditions were optimized with a checkerboard serial-dilution analysis. The HRP-conjugated antibody dilutions tested were 1:2000, 1:4000, 1:5000, and 1:10,000, under the set incubation conditions described above. The conditions that produced the largest positive/negative ratio (P/N) of OD_450_ (OD_450_ of positive serum/OD_450_ of negative serum) were deemed the best reaction conditions.

Then, 100 sera samples which were randomly selected from 133 sera were tested simultaneously. The sample/positive control (S/P) values were calculated as: (sample OD_450_, negative control OD_450_)/(positive control OD_450_). The cut-off value was calculated with a receiver operating characteristic (ROC) analysis.

### 2.4. Specificity Analysis

A total of 35 specific sera samples of different viruses (GETV-P6~P10,GETV-N4~N8,JE-P1~P5,PRRS-P1~P5,CSF-P1~P5,PR-P1~P5,PED-P1~P5, Appendix A) were used to evaluate the specificity of the ELISA.

### 2.5. Sensitivity Analysis

GETV serum (GETV-P1~P3, GETV-P4~P6, GETV-P7~P9, GETV-N8~N10, Appendix A) was diluted 1:100, 1:200, 1:400, 1:800, 1:1600, 1:3200, 1:6400, 1:12,800, 1:25,600, or 1:51,200, and analyzed with the established ELISA to determine its limit of detection.

### 2.6. Reproducibility Assay for the Indirect ELISA

The reproducibility of the ELISA was evaluated with six serum samples. The coefficient of variation (CV) was used to evaluate the intra- and interassay variation. Each sample was tested on three plates on different occasions to determine the interassay CV, and three replicates within the same plate were used to calculate the intra-assay CV. The mean S/P ratios and standard deviations (SD) were also calculated.

### 2.7. Coincidence Test of the Indirect ELISA

A total of 133 clinical sera samples from pig farms were tested with the ELISA developed here, and the results were compared with those of IFA to determine the coincidence rate.

### 2.8. Statistical Analysis

All results are presented as the means ± standard errors of the means (SE) of triplicate experiments. The data were analyzed with GraphPad Prism 8.4.3 (GraphPad Software, San Diego, CA, USA) and SPSS 22.0 (IBM Corp, Armonk, NY, USA). Statistical significance was evaluated with one-way analysis of variance (ANOVA).

## 3. Results

### 3.1. Expression and Purification of rE2 Protein

The rE2 protein was purified by elution against an imidazole gradient, and yielded a purified protein concentration of 1.96 μg/μL. An SDS-PAGE analysis demonstrated that the rE2 protein had an approximate molecular mass of 46 kDa (Figure 1A: lanes 4 and 5). However, Western blotting with an anti-His monoclonal antibody and known GETV-positive pig serum showed the shifted band to be the rE2 protein with the monoclonal antibody (Figure 1B,C)

### 3.2. ELISA Optimization with rE2 Protein

The conditions for antigen coating were optimized at different temperatures and times. The optimal ELISA result was obtained when the wells were coated with antigen at 4 °C for 12 h. The optimal working concentration of antigen was 24.5 μg/mL and the appropriate serum dilution was 1:100. These conditions were determined with checkerboard assays using serial dilutions of the antigen and sera (Table 1).

The optimum conditions were antigen coating at 4 °C for 12 h, blocking at 37 °C for 1 h, and incubation with GETV-positive serum at 37 °C for 1 h. The HRP-conjugated goat anti-pig IgG antibody was optimally diluted 1:5000, and the best reaction conditions were 37 °C for 1 h. All these results were based on the principle that the P/N ratio should be >2.1 [22].

### 3.3. ROC Curve Analysis

One hundred pig serum samples collected in the field were analyzed with IFA, and the results of this analysis were compared with those of the ELISA developed in this study based on an rE2 protein. Using analyze -roc curve in SPSS 22.0, the ROC analysis showed that the area under the curve (AUC) for the ELISA was 0.964 (95% confidence interval [CI], 0.940−0.989; Figure 2A), and the sensitivity and specificity were 97.1% and 93.3%, respectively. The cut-off value was determined to be 0.344 [23,24].

### 3.4. Specificity Analysis of the Indirect ELISA

To test the cross-reactivity of the rE2-based indirect ELISA, anti-sera from other common porcine viruses including JEV, PRRSV, CSFV, PEDV, and PRV were examined. The average resultant S/P of the JEV (S/P = 0.220), PRRSV (S/P = 0.152), CSFV (S/P = 0.137), PEDV (S/P = 0.220), and PRV (S/P = 0.227) anti-sera were lower than the positive cut-off value (0.344), confirming that the established rE2 indirect ELISA was non-cross-reactive with these samples. Thus, the indirect ELISA proved to possess high specificity (Figure 2B).

### 3.5. Sensitivity Analysis Test of the Indirect ELISA

Different dilutions of positive serum were tested. The results showed that the limit of detection with this method was a dilution of 1:12,800 (Figure 2C).

### 3.6. Reproducibility of the Indirect ELISA

Six pig sera were tested three times in duplicate using three established batches of ELISA-coated plates. The results showed that the intra-assay CV of this method was 2.25–6.16% and the interassay CV was 3.29–5.81%, so both were <7% (Table 2), indicating that the method has a high degree of repeatability. It is therefore highly accurate and can be used for the routine detection of GETV.

### 3.7. Coincidence of the Indirect ELISA

When 133 samples were tested with ELISA and IFA, the ELISA identified three false-positive samples and four false-negative samples (Table 3). Therefore, the coincidence rate of detection with these two methods was 94.74% (126/133).

### 3.8. Relationship between Swine GETV-Antibody-Positive Rate and Province in Eastern China

Antibody testing of pig sera from Eastern China (Shandong, Hebei, Zhejiang, Shanghai, Jiangsu, and Guangdong Provinces) showed that the positive rates in Shandong, Hebei, and Shanghai were <30.00%, <35.00%, and <35.00%, respectively, whereas the positive rates in Zhejiang, Jiangsu and Guangdong were 45.00%, 43.48%, and 36.67%, respectively (Table 4). However, the differences among the provinces were not significant (*p* > 0.05).

### 3.9. Relationship between Swine GETV-Antibody-Positive Rate and Season in Eastern China

October is the breeding season for mosquitoes in Eastern China, and it is also the epidemic period for GETV. A statistical analysis of the numbers of GETV-antibody-positive samples in April and October showed that the average positivity rate in spring (April) was 24.24%, which was significantly lower than the average positivity rate in autumn (October, 50.75%), after the epidemic season. The difference was significant (*p* < 0.05; Table 4), indicating that the rate of GETV antibody positivity in pigs in Eastern China has a certain seasonality.

## 4. Discussion

GETV is mainly distributed in southern cities of China [12,13,25], and the epidemic period extends from July to September each year, coinciding with the period of high mosquito incidence. GETV causes death in piglets and abortion in sows, resulting in economic losses, and detection methods for this virus must be established. The alphavirus E2 glycoprotein is a surface protein and is highly immunogenic. It has three functional domains (A, B, and C). Domains A and B are exposed on the viral membrane and are responsible for viral attachment to the host cell [15,26,27]. Because these domains contain antigenic epitopes for virus-neutralizing antibodies, they have been used as targets for the development of serodiagnostic tests for *Chikungunya virus* [28,29].

Virus isolation and identification is the most commonly used pathogen detection method, but is complicated and time-consuming [30]. PCR and real-time PCR instruments are expensive and difficult to use. ELISAs have the advantages of simple operation, high sensitivity, and suitability for large-scale sample identification, and they are commonly used in laboratories. In the present study, we improved an ELISA based on the rE2 protein of GETV. A ROC curve analysis of the GETV-E2 ELISA results showed that its sensitivity was 97.1% and its specificity 93.3%. The AUC was 0.964, indicating a high level of diagnostic accuracy. This was confirmed in a reproducibility test that showed that all CVs (for intra-assay, interassay, and interlaboratory comparisons) were <7%. No cross-reactivity was observed with antisera against PEDV, JEV, CSFV, PRV, or PRRSV.

These results suggested that we re-confirmed the usefulness of the rE2-ELISA, providing appropriate references for both the diagnosis of GETV infection and seroepidemiological surveys. In 2017–2018, the rate of GETV infection in pigs in Thailand was 23.1%, and two different GETVs were circulating [7]. GETV was detected in Foshan City, Guangdong Province, China in 2018, and was very similar (99.7%) to swine virus strain HNJZ-S2 in Henan [25]. Therefore, GETV has been detected in various countries, so a reliable test is required.

When 133 serum specimens collected from six provinces in Eastern China in 2018 were clinically tested, 50 (37.59%) of the specimens tested positive for anti-GETV IgG antibodies with the GETV-E2 ELISA. Zhejiang Province had the highest positivity rate of 45.00%. The positivity rates for Zhejiang, Jiangsu, and Guangdong Provinces were higher than those for Shandong, Hebei, and Shanghai Provinces. Therefore, GETV may spread more easily in southern China than in the north because the climate in the south is humid, with, consequently, more mosquitoes.

In October, China has a high temperature, high humidity, and high mosquito density. The seroprevalence of GETV was significantly lower in April (24.24%, 16/66) than in October (50.75%, 34/67). The rate of GETV antibody positivity is related to the epidemic season of GETV, so the positive rate of GETV antibodies in autumn (October) after the mosquito epidemic season was significantly higher than in April, before the epidemic season, which means that the incidence of GETV was related to the season.

Our data indicated that the prevalence of GETV was relatively high in Eastern China. Further research on the circulation of GETV in China will extend our understanding of the epidemiology of the disease associated with this virus.

## Figures and Tables

**Figure 1 viruses-14-02173-f001:**
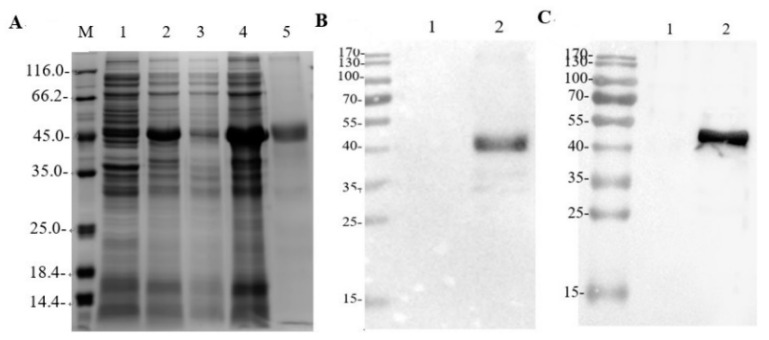
Expression and purification of recombinant E2 protein. (**A**) SDS-PAGE analysis of E2 protein expressed in *Escherichia coli*. The E2 protein was 46 kDa on SDS-PAGE. Expression of pColdI–E2 in *E. coli* was induced with isopropyl β-d-1-thiogalactopyranoside (IPTG). Inclusion bodies were collected 16 h after induction and subjected to supersonic schizolysis. M, marker; lane 1: uninduced *E. coli* containing pColdI–E2; 2: *E. coli* containing pColdI–E2 induced with IPTG; 3: supernatant of IPTG-induced *E. coli* containing pColdI–E2; 4: inclusion bodies from IPTG-induced *E. coli* containing pColdI–E2; 5: purified *E. coli* containing pColdI–E2. (**B**) Western blotting analysis of E2 protein with an anti-His antibody. Uninduced *E. coli* containing pColdI–E2 and E2 protein were resolved electrophoretically on 12% polyacrylamide gel and transferred to 0.2 μm polyvinylidene difluoride membrane. Membranes were treated with an anti-His antibody followed by HRP-conjugated goat anti-mouse IgG antibody. Lane 1: uninduced *E. coli* containing pColdI–E2; 2: E2 protein. (**C**) Reactivity of serum from GETV-infected pigs with the viral antigen. Uninduced *E. coli* containing pColdI–E2 and E2 protein were resolved electrophoretically on 12% polyacrylamide gel and transferred to 0.2 μm polyvinylidene difluoride membrane. The membranes were treated with pig sera (GETV-P1) followed by HRP-conjugated goat anti-pig IgG antibody. The reaction was visualized with ECL Western Blotting Substrate. Lane 1: uninduced *E. coli* containing pColdI–E2; 2: E2 protein.

**Figure 2 viruses-14-02173-f002:**
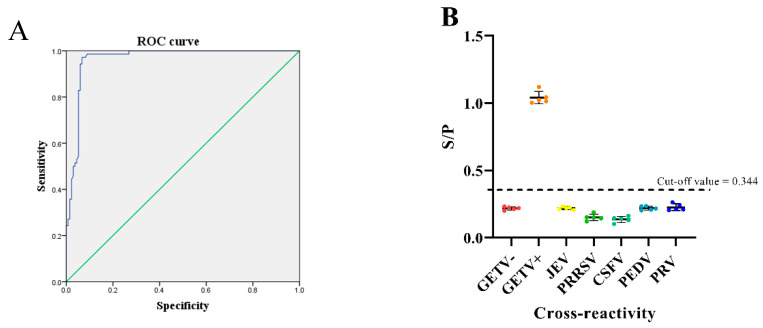
Validation of the indirect ELISA. (**A**) Receiver operating characteristic (ROC) analysis of the developed ELISA. The blue line represents the test curve and the green line corresponds to the noninformative test curve. The sensitivity and specificity of the developed ELISA were 97.1% and 93.3%, respectively, when the optimal cut-off OD_450_ value was 0.344. The area under the ROC curve (AUC) of the developed ELISA was 0.964 (95% CI, 0.940–0.989). (**B**) Specificity analysis. The average resultant OD_450_ of the JEV, PRRSV, CSFV, PEDV, PRV anti-sera and GETV-negative sera were lower than the positive cut-off value, and only the OD_450_ of the GETV-positive sera was higher than the positive cut-off value. (**C**) Sensitivity analysis. GETV-positive serum was serially diluted twofold (from 1:100 to 1:51,200) and tested with the ELISA. The maximum detectable dilution was 1:12,800. The positive cut-off value (0.344) for the ELISA is indicated with dashed lines. The values below the line (cut of value) were considered negative.

**Table 1 viruses-14-02173-t001:** Determination of the optimal protein coating concentration and serum dilution.

Serum Dilution		Protein Coating Concentration (μg/mL)						
		196.000	98.000	49.000	24.500 *	12.250	6.130	3.030
1:25	P	2.871 ± 0.021	2.870 ± 0.028	2.651 ± 0.019	2.885 ± 0.061	2.716 ± 0.068	2.823 ± 0.027	2.863 ± 0.068
	N	0.279 ± 0.006	0.287 ± 0.003	0.273 ± 0.004	0.302 ± 0.008	0.288 ± 0.002	0.379 ± 0.004	0.463 ± 0.006
	P/N	10.290	10.004	9.710	9.568	9.418	7.456	6.189
1:50	P	2.546 ± 0.043	2.432 ± 0.033	2.111 ± 0.011	2.064 ± 0.159	1.886 ± 0.086	1.774 ± 0.023	1.712 ± 0.024
	N	0.173 ± 0.005	0.175 ± 0.004	0.181 ± 0.003	0.183 ± 0.003	0.194 ± 0.001	0.198 ± 0.002	0.241 ± 0.003
	P/N	14.717	13.897	11.663	11.279	9.722	8.966	7.106
1:100 *	P	1.871 ± 0.069	1.870 ± 0.034	1.651 ± 0.047	1.885 ± 0.031	1.716 ± 0.032	1.823±0.057	1.863 ± 0.022
	N	0.086 ± 0.012	0.091 ± 0.003	0.087 ± 0.005	0.073 ± 0.001	0.098 ± 0.011	0.130 ± 0.009	0.143 ± 0.002
	P/N	21.684	20.605	19.013	25.822 *	17.579	14.043	13.000
1:200	P	1.704 ± 0.064	1.741 ± 0.032	1.555 ± 0.052	1.726 ± 0.017	1.579 ± 0.056	1.774 ± 0.024	1.725 ± 0.011
	N	0.083 ± 0.012	0.088 ± 0.010	0.082 ± 0.009	0.108 ± 0.004	0.111 ± 0.006	0.144 ± 0.007	0.142 ± 0.005
	P/N	20.421	19.871	18.868	16.014	14.221	12.343	12.115

P: positive serum; N: negative serum. Optimal conditions selected the maximum P/N value. The OD values presented are average values. “*” means the optimum conditions.

**Table 2 viruses-14-02173-t002:** Repeatability of the indirect ELISA.

Sample Number	Interassay CV (%)		Intra-Assay CV(%)	
	X ± SD	CV(%)	X ± SD	CV(%)
1	1.147 ± 0.038	3.29%	1.140 ± 0.026	2.25%
2	0.556 ± 0.032	5.78%	0.554 ± 0.033	5.94%
3	0.775 ± 0.029	3.79%	0.785 ± 0.048	6.16%
4	0.852 ± 0.043	5.06%	0.857 ± 0.050	5.84%
5	0.471 ± 0.019	3.99%	0.471 ± 0.017	3.66%
6	1.203 ± 0.070	5.81%	1.195 ± 0.060	5.05%

**Table 3 viruses-14-02173-t003:** Results of coincidence testing.

		IFA Results		
		Positive	Negative	Total
Indirect ELISA results	Positive	47	4	51
	Negative	3	79	82
	Total	50	83	133

**Table 4 viruses-14-02173-t004:** Pig GETV antibody test results.

Variable	Category	No. Examined	No. Positive	Positive Rate	*p*-Value
Provinces	Shandong	20	6	30.00%	0.24
	Hebei	20	7	35.00%	0.24
	Zhejiang	20	9	45.00%	0.19
	Shanghai	20	7	35.00%	0.86
	Jiangsu	23	10	43.48%	0.73
	Guangdong	30	11	36.67%	0.43
Season	Spring	66	16	24.24%	0.03
	Autumn	67	34	50.75%	0.04
Total		133	50	37.59%	

## Data Availability

Not applicable.

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
