# Peer review of "Seroprevalence of Getah virus in Pigs in Eastern China Determined with a Recombinant E2 Protein-Based Indirect ELISA"

_viruses, 2022, doi:10.3390/v14102173_

Round 1

Reviewer 1 Report (Previous Reviewer 2)

The resubmitted version is improved and suitable for publication in viruses

Author Response

Dear Editors and Reviews:

Thank you for your letter and for the reviews’ comments concerning our manuscript entitled “Seroprevalence of Getah virus in pigs in Eastern China determined with a recombinant-E2-protein-based indirect ELISA” (ID: viruses-1954936). We appreciate your recommendation and wish you to have a happy life.

Reviewer 2 Report (Previous Reviewer 1)

Comments to the manuscript ID: viruses-1954936

Re the response to my comment #1,

In this submission, the authors provided two additional tables (Supplementary Table S1 and S2) showing the infection status of “naturally-infected pigs” with the results of RT-PCR, which would fulfill the requirement that I had pointed. However, they did not add this important information in the main document, and readers still cannot follow. They have to add the following in the materials and methods section: “For these 20 sera in Suppl Table S1, GETV-infected/non-infected status was identified by VN, IFN and a real-time RT-PCR tests” with the appropriate reference paper (Zhang et al., 2022).

 Re the response to my comment #4 and 6

The authors provided information on which specific sera in Suppl Table S1 and S2 were used in each experiment. However, this has also not been included in the main document, and readers still cannot follow. I found “Supplementary Table S1” in line 83, but could not find “Supplementary Table S2” anywhere in the main document. Also, regarding the Fig. 2C, it seemed that they used three sets of pooled sera (each consisted with three pigs) for positives, and one set of pooled sera consisted with three pigs for negatives. But this information is also not included in the materials and methods, and readers cannot follow. Furthermore, in the response letter, they described “GETV positive sera (1,2,3,4,5) and negative sera (1,2,3,4,5)…”, but there’s no negative sera which were identified as #1 to #5 in the Suppl Table S1, as the negative sera should be from #11 to #20.

 Re the response to my comment #5

A contradiction between the response to comment #5 and the main document. In the response letter, they wrote, “…for the test (Table 1), GETV positive sera (4,5,6) and negative sera (4,5,6) experiment were all from the sera in Table S1”. So they used three positive sera and three negative sera. However, in the main document (line 96-97), it was written that “One positive serum and one negative serum were used to optimize ELISA.” Which information is correct? And again, there’s no negative sera identified as #4 to #6 in the Suppl Table S1.

 Overall, the authors must provide sufficient information not only to the reviewers but also to the potential readers. Just writing explanation in the response letter does not make any sense, and they have to add description in the paper, so that the readers can understand what the authors meant. Also, there’s still too much contradiction, and the authors should pay much attention in amending the manuscript.

Author Response

Dear Editors and Reviews:

Thank you for your letter and for the reviews’ comments concerning our manuscript entitled “Seroprevalence of Getah virus in pigs in Eastern China determined with a recombinant-E2-protein-based indirect ELISA” (ID: viruses-1954936). Those comments are all valuable and very helpful for revising and improving our paper, as well as the important guiding significance to researches.

We have studied comments carefully and have made correction which we hope meet with approval. Revised portion are marked in red in the paper. The main corrections in the paper and the responds to the reviewer’s comments are as flowing:

1  In this submission, the authors provided two additional tables (Supplementary Table S1 and S2) showing the infection status of “naturally-infected pigs” with the results of RT-PCR, which would fulfill the requirement that I had pointed. However, they did not add this important information in the main document, and readers still cannot follow. They have to add the following in the materials and methods section: “For these 20 sera in Suppl Table S1, GETV-infected/non-infected status was identified by VN, IFN and a real-time RT-PCR tests” with the appropriate reference paper (Zhang et al., 2022).

Response:

We are grateful for the suggestion. To be more clearly and in accordance with the reviewer concerns, we have added a more detailed interpretation regarding serum. We supplement the background introduction of these 20 serums. We changed into “20 sera (GETV-P1~P10 and GETV-N1~N10) from naturally infected pigs, which were tested by VN [13], IFA [31], qRT-PCR [32], were used to establish and optimize the ELISA protocol (Supplementary table S1). ” (lines 99-101).

  1. 1  Li, Y.; Fu, S.; Guo, X.; Li, X.; Li, M.; Wang, L.; Liang, G. Serological survey of Getah virus in domestic animals in Yunnan province, China. Vector-Borne and Zoonotic Diseases. 2019, 19, 59–61.
  2.  Hameed M, Wahaab A, Shan T, Wang X, Khan S, Di D, Xiqian L, Zhang JJ, Anwar MN, Nawaz M, Li B, Liu K, Shao D, Qiu Y, Wei J, Ma Z. A Metagenomic Analysis of Mosquito Virome Collected From Different Animal Farms at Yunnan-Myanmar Border of China. Front Microbiol. 2021,11,591478.
  3.  Zhang Y, Li Y, Guan Z, Yang Y, Zhang J, Sun Q, Li B, Qiu Y, Liu K, Shao D, Ma Z, Wei J, Li P. Rapid Differential Detection of Japanese Encephalitis Virus and Getah Virus in Pigs or Mosquitos by a Duplex TaqMan Real-Time RT-PCR Assay. Front Vet Sci. 2022,9,839443.

2  The authors provided information on which specific sera in Suppl Table S1 and S2 were used in each experiment. However, this has also not been included in the main document, and readers still cannot follow.

Response:

Thank you for your suggestion. We found the referee’s comments most helpful and have revised the manuscript. 

We changed into “20 sera (GETV-P1~P10 and GETV-N1~N10) from naturally infected pigs, which were tested by VN [13], IFA [31], qRT-PCR [32], were used to establish and optimize the ELISA protocol (Supplementary table S1).” (lines 99-101).

“Positive sera of Japanese encephalitis virus (JEV,JE-P1~P5), Porcine Reproductive and Respiratory Syndrome(PRRSV,PRRS-P1~P5), Classical swine fever virus(CSFV,CSF-P1~P5), Pseudorabies virus (PRV,PR-p1~p5), and Porcine epidemic diarrhea virus(PEDV,PED-P1~P5) verified by VN and IFA (Supplementary table S2) were from experimentally-infected pigs and provided by the China Animal Health and Epidemiology Center (Shanghai Branch) and was used to optimize ELISA protocols.”(lines 101-107).

“The recombinant vector was confirmed with SDS-PAGE and the expressed protein with a Western blotting assay based on GETV-positive serum (GETV-P1) and a His-tagged antibody.” (lines 113-115)

“An indirect ELISA was carried out and optimized with positive (GETV-P2~P4) and negative (GETV-N1~N3) controls serum samples.” (lines 119-120)

“35 specific sera of different viruses (GETV-P6~P10,GETV-N4~N8,JE-P1~P5,PRRS-P1~P5,CSF-P1~P5,PR-P1~P5,PED- P1~P5, Supplementary table S1 or S2) were used to evaluate the specificity of the ELISA.” (lines 148-150)

“GETV serum(GETV-P1~P3, GETV-P4~P6, GETV-P7~P9, GETV-N8~N10, Supplementary table S1) was diluted 1:100, 1:200, 1:400, 1:800, 1:1,600, 1:3,200, 1:6,400, 1:12,800, 1:25,600, or 1:51,200, and analyzed with the established ELISA to determine its limit of detection.” (lines 152-155)

3  I found “Supplementary Table S1” in line 83, but could not find “Supplementary Table S2” anywhere in the main document.

Response:

Thank you for your suggestion. We found the referee’s comments most helpful and have revised the manuscript. We changed into “Positive sera of Japanese encephalitis virus (JEV,JE-P1~P5), Porcine Reproductive and Respiratory Syndrome(PRRSV,PRRS-P1~P5), Classical swine fever virus(CSFV,CSF-P1~P5), Pseudorabies virus (PRV,PR-p1~p5), and Porcine epidemic diarrhea virus(PEDV,PED-P1~P5) verified by VN and IFA (Supplementary table S2) were from experimentally-infected pigs and provided by the China Animal Health and Epidemiology Center (Shanghai Branch) and was used to optimize ELISA protocols.” (lines 101-107)

4  Also, regarding the Fig. 2C, it seemed that they used three sets of pooled sera (each consisted with three pigs) for positives, and one set of pooled sera consisted with three pigs for negatives. But this information is also not included in the materials and methods, and readers cannot follow.

Response:

Thank you for your suggestion. We found the referee’s comments most helpful and have revised the manuscript. Modifications are as follows. We changed into “GETV serum(GETV-P1~P3, GETV-P4~P6, GETV-P7~P9, GETV-N8~N10, Supplementary table S1) was diluted 1:100, 1:200, 1:400, 1:800, 1:1,600, 1:3,200, 1:6,400, 1:12,800, 1:25,600, or 1:51,200, and analyzed with the established ELISA to determine its limit of detection.” (lines 152-155)

5  Furthermore, in the response letter, they described “GETV positive sera (1,2,3,4,5) and negative sera (1,2,3,4,5)…”, but there’s no negative sera which were identified as #1 to #5 in the Suppl Table S1, as the negative sera should be from #11 to #20.

Response:

Thank you for your suggestion. We found the referee’s comments most helpful and have revised the manuscript. Modifications are as follows. First, We changed the Supplementary table S1. GETV P1-P10 was positive serum and N1-N10 was negative. Second, we have changed “35 specific sera of different viruses (GETV-P6~P10,GETV-N4~N8,JE-P1~P5,PRRS-P1~P5,CSF-P1~P5,PR-P1~P5,PED- P1~P5, Supplementary table S1 or S2) were used to evaluate the specificity of the ELISA.”(lines 148-150)

6  A contradiction between the response to comment #5 and the main document. In the response letter, they wrote, “…for the test (Table 1), GETV positive sera (4,5,6) and negative sera (4,5,6) experiment were all from the sera in Table S1”. So they used three positive sera and three negative sera. However, in the main document (line 96-97), it was written that “One positive serum and one negative serum were used to optimize ELISA.” Which information is correct?

Response:

Thank you for your suggestion. We found the referee’s comments most helpful and have revised the manuscript. Modifications are as follows. We changed into “An indirect ELISA was carried out and optimized with positive (GETV-P2~P4) and negative (GETV-N1~N3) controls serum samples. ” (lines 119-120)

7  And again, there’s no negative sera identified as #4 to #6 in the Suppl Table S1.

Response:

Thank you for your suggestion. We found the referee’s comments most helpful and have revised the manuscript. Modifications are as follows. We have changed the Supplementary table S1. GETV-P1~P10 were positive serum and GETV-N1~N10 were negative. 

This manuscript is a resubmission of an earlier submission. The following is a list of the peer review reports and author responses from that submission.

Round 1

Reviewer 1 Report

Comments for the manuscript viruses-1769989

In this paper, the authors described an ELISA for the detection of antibodies to Getah virus (GETV) in pigs using recombinant E2 (rE2) glycoprotein as an antigen. They showed optimal experimental settings for this ELISA using positive and negative control sera as well as field sera. Also, they performed serosurveillance for pig sera collected in various locations in China, and found substantial prevalence of GETV with seasonal difference. However, there are serious concerns to be solved before publication.

First, the rE2-ELISA for detection of GETV infection in horses had been published several years ago (Bannai et al., 2019, cited as ref. 18), and its application to pig sera have also been published by the other group (Shi et al., 2022. Virol Sinica 37:229-237). So, the usefulness of rE2 protein in detecting GETV-antibodies was not their original finding, and they should state that the current study was aimed to determine the optimal experimental settings of rE2-ELISA (which has already been developed by the other groups) for its use in pig sera. Otherwise, readers will misunderstand that the authors of this paper “developed” the rE2-ELISA.

Second, the authors compared the results of rE2-ELISA with immunofluorescent assay (IFA), which has not been validated for its diagnostic accuracy for GETV. A virus-neutralization (VN) test is currently most reliable method (gold standard) for GETV serology (Fukunaga et al., 2000. Vet Clin North Am Equine Prac 16:605-617), and many reports on GETV ELISAs have been validated in comparison with the results from the VN test (Kuwata et al., 2018. Arch Virol 163:2817-2821; Lee et al., 2016. J Bacteriol Virol 46:63-70; Bannai et al., 2019 and 2020 as ref. 18-19). Therefore, the authors should perform VN test instead of IFA to validate the sensitivity, specificity and coincidence of the rE2-ELISA.

Finally, this paper lacks important information about the origin of serum samples in many parts (see the comments below). Certainty of control sera in terms of proof of “infected” or “non-infected” status is essential in establishing or determining experimental settings of serological tests. The authors should take much care about description of serum origins, so that they can convince the readers.

L14-16. “To develop a high-throughput, time efficient method…”. Overstatement. Immunogenic antigens of GETV have already been investigated, and the rE2-ELISA has already been developed by the other groups. The objective of this study should be optimization of experimental conditions of rE2-ELISA for pig sera, and its application for a serosurveillance.

L28. “We have established an rE2-ELISA…”. Overstatement. The authors just showed optimal settings of rE2-ELISA for its use for pig sera.

L74-78. There was no description for the details of positive and negative control sera: whether they were from experimentally-infected pigs or naturally-infected pigs; whether they were taken from one pig or pooled sera from multiple pigs; VN titers, etc. If they were collected from naturally-infected pigs, the concrete evidence of GETV infection in these animals should be described (i.e. RT-PCR, virus-isolation, etc.).

L79-85. The strain name of GETV used for the cloning of E2 gene and its RNA sequence should be disclosed.

L114-116. The authors should describe the details of positive control sera for GETV, JEV, PEDV, PRV, CSFV and PRRSV (i.e. experimentally-infected pigs or naturally-infected pigs), with the concrete evidence of infection with these viruses (i.e. RT-PCR, virus-isolation, etc.).

L110, L249-250. Lack of information again, for the 100 field sera used for the ROC analysis (geographical location, single farm or multiple farms, age, clinical status, etc.). Are they distinct from the 133 sera tested in Tables 3 and 4? The authors should describe the details in Materials and Methods section. Moreover, the seropositivity/negativity was determined by the Western blot analysis, whose diagnostic accuracy has also not been validated. The authors should add the result of VN test for these sera, and evaluate the sensitivity, specificity and coincidence with the rE2-ELISA.

L170-171. The authors should provide a reference for the principle that “the P/N ratio should be over 2.1”.

L201-203. This experiment (Fig. 3C) does not make any sense. I suggest to remove this.

L211-214 (Table 3). Very confusing. If it is a part of validation of rE2-ELISA, the authors should indicate the sensitivity and specificity as well. Or, if it is a part of serosurveillance, we can get no information from the coincidence of rE2-ELISA and IFA. Overall, the use of two sets of field sera for validation of ELISA (100 sera in Fig. 2A and 133 sera in Table 3) is irrational.

L234. “GETV is mainly distributed in southern cities of China” The authors should provide references.

L247-248. “We established an ELISA based on the rE2 protein of GETV”. Overstatement.

L255. “newly developed GETV-E2 ELISA”. This ELISA was developed by the other groups and is not the original achievement of this paper. The authors should state that they re-confirmed the usefulness of the rE2-ELISA, providing appropriate references.

Author Response

Dear Editors and Reviews:

Thank you for your letter and for the reviews’ comments concerning our manuscript entitled “Seroprevalence of Getah virus in pigs in Eastern China determined with a recombinant-E2-protein-based indirect ELISA” (ID: viruses-1769989). Those comments are all valuable and very helpful for revising and improving our paper, as well as the important guiding significance to researches. We have studied comments carefully and have made correction which we hope meet with approval. Revised portion are marked in red in the paper. The main corrections in the paper and the responds to the reviewer’s comments are as flowing:

Reviewer 1

In this paper, the authors described an ELISA for the detection of antibodies to Getah virus (GETV) in pigs using recombinant E2 (rE2) glycoprotein as an antigen. They showed optimal experimental settings for this ELISA using positive and negative control sera as well as field sera. Also, they performed serosurveillance for pig sera collected in various locations in China, and found substantial prevalence of GETV with seasonal difference. However, there are serious concerns to be solved before publication.

First, the rE2-ELISA for detection of GETV infection in horses had been published several years ago (Bannai et al., 2019, cited as ref. 18), and its application to pig sera have also been published by the other group (Shi et al., 2022. Virol Sinica 37:229-237). So, the usefulness of rE2 protein in detecting GETV-antibodies was not their original finding, and they should state that the current study was aimed to determine the optimal experimental settings of rE2-ELISA (which has already been developed by the other groups) for its use in pig sera. Otherwise, readers will misunderstand that the authors of this paper “developed” the rE2-ELISA.

Authors’ response: Thanks for the professional comment. I have modified it as per your comments and changed it to " Therefore, the experimental work of GETV detection is very necessary. However, due to the influence of a variety of unavoidable factors, the ELISA test for the primary screening of animal diseases has low accuracy of the detection results. Therefore, we optimized an enzyme-linked immunosorbent assay (ELISA) based on recombinant E2 (rE2) protein (lines 28-33). In the present study, we improved an ELISA based on the rE2 protein of GETV (lines 302-303). "

Second, the authors compared the results of rE2-ELISA with immunofluorescent assay (IFA), which has not been validated for its diagnostic accuracy for GETV.A virus-neutralization (VN) test is currently most reliable method (gold standard) for GETV serology (Fukunaga et al., 2000. Vet Clin North Am Equine Prac 16:605-617), and many reports on GETV ELISAs have been validated in comparison with the results from the VN test (Kuwata et al., 2018. Arch Virol 163:2817-2821; Lee et al., 2016. J Bacteriol Virol 46:63-70; Bannai et al., 2019 and 2020 as ref. 18-19). Therefore, the authors should perform VN test instead of IFA to validate the sensitivity, specificity and coincidence of the rE2-ELISA.

Authors’ response:Thank you for your valuable suggestion.Indeed,the virus-neutralization (VN) test is the gold standard for GETV antibody detection. According to our previous report, immunofluorescent assay (IFA)can also be used for the detection of GETV antibodies(Hameed M,et al. 2021).According to suggestions,we selected 20 serum samples, including the serum used to verify the sensitivity and specificity of the rE2-ELISA and partial serum from the coincidence of the rE2-ELISA, and tested them by VN test and IFA test in order to compare the consistency of the two methods,The results show that the two methods are consistent in detecting GETV antibody(Supplementary table S1). We have corrected this comment in our revised manuscript (lines 101-104).

Reference:

31  Hameed M, Wahaab A, Shan T, Wang X, Khan S, Di D, Xiqian L, Zhang JJ, Anwar MN, Nawaz M, Li B, Liu K, Shao D, Qiu Y, Wei J, Ma Z. A Metagenomic Analysis of Mosquito Virome Collected From Different Animal Farms at Yunnan-Myanmar Border of China. Front Microbiol. 2021 Feb 8;11:591478. doi: 10.3389/fmicb.2020.591478. PMID: 33628201; PMCID: PMC7898981.

Finally, this paper lacks important information about the origin of serum samples in many parts (see the comments below). Certainty of control sera in terms of proof of “infected” or “non-infected” status is essential in establishing or determining experimental settings of serological tests. The authors should take much care about description of serum origins, so that they can convince the readers.

Authors’ response: Thank you for your suggestion. We found the referee’s comments most helpful and have revised the manuscript. Modifications are as follows. A total of 133 clinical pig serum samples were collected from finishing pigs(24– 28 weeks old) in slaughterhouses or sow (more than 28 weeks old) on pig farms in six provinces (Shandong,Hebei, Zhejiang, Shanghai, Jiangsu, Guangdong) in Eastern China, in April (n =66)and in October (n =67) of 2018. Sample information is given in Table 4. JEV (Japanese encephalitis virus), PRRSV (Porcine Reproductive and Respiratory Syndrome), CSFV (Classical swine fever virus), and PRV (Pseudorabies virus), PEDV (porcine epidemic diarrhea virus) positive serum was from experimentally-infected pigs and provided by the China Animal Health and Epidemiology Center (Shanghai Branch). 20 GETV positive or negative sera collected from domestic pigs were screened and validated by the virus-neutralization (VN) test [13] or IFA [31] for optimization , specificity, sensitivity of the ELISA protocol (Supplementary table S1). We have corrected this comment in our revised manuscript (lines 93-104).

L14-16. “To develop a high-throughput, time efficient method…”. Overstatement. Immunogenic antigens of GETV have already been investigated, and the rE2-ELISA has already been developed by the other groups. The objective of this study should be optimization of experimental conditions of rE2-ELISA for pig sera, and its application for a serosurveillance.

Authors’ response:Thank you for your suggestion. We found the referee’s comments most helpful and have revised the manuscript. In the revised manuscript, we changed it to " Therefore, the experimental work of GETV detection is very necessary. However, due to the influence of a variety of unavoidable factors, the ELISA test for the primary screening of animal diseases has low accuracy of the detection results. Therefore, we optimized an enzyme-linked immunosorbent assay (ELISA) based on recombinant E2 (rE2) protein (lines 28-33). In the present study, we improved an ELISA based on the rE2 protein of GETV (lines 306-307). "

L28. “We have established an rE2-ELISA…”. Overstatement. The authors just showed optimal settings of rE2-ELISA for its use for pig sera.

Authors’ response:Thank you for your suggestion. We found the referee’s comments most helpful and have revised the manuscript. In the revised manuscript, we changed it to “In conclusion, we have improved an rE2 ELISA that detected pig antibodies against GETV after experimental and natural infections. (lines 47-48)”

L74-78. There was no description for the details of positive and negative control sera: whether they were from experimentally-infected pigs or naturally-infected pigs; whether they were taken from one pig or pooled sera from multiple pigs; VN titers, etc. If they were collected from naturally-infected pigs, the concrete evidence of GETV infection in these animals should be described (i.e. RT-PCR, virus-isolation, etc.).

Authors’ response:Thank you for your suggestion. We found the referee’s comments most helpful and have revised the manuscript. Modifications are as follows: A total of 133 clinical pig serum samples were collected from finishing pigs(24– 28 weeks old) in slaughterhouses or sow (more than 28 weeks old) on pig farms in six provinces (Shandong,Hebei, Zhejiang, Shanghai, Jiangsu, Guangdong) in Eastern China, in April (n =66)and in October (n =67) of 2018. Sample information is given in Table 4. JEV (Japanese encephalitis virus), PRRSV (Porcine Reproductive and Respiratory Syndrome), CSFV (Classical swine fever virus), and PRV (Pseudorabies virus), PEDV (porcine epidemic diarrhea virus) positive serum was from experimentally-infected pigs and provided by the China Animal Health and Epidemiology Center (Shanghai Branch). 20 GETV positive or negative sera collected from domestic pigs were screened and validated by the virus-neutralization (VN) test [13] or IFA [31] for optimization , specificity, sensitivity of the ELISA protocol (Supplementary table S1). We have corrected this comment in our revised manuscript (lines 93-104).

L79-85. The strain name of GETV used for the cloning of E2 gene and its RNA sequence should be disclosed.

Authors’ response: Thank you for your suggestion. We found the referee’s comments most helpful and have revised the manuscript. Modifications are as follows. We refer to the complete gene sequence of structural protein E2 of GETV (SH05-6) in GenBank (EU015066), whose length is 1266 bp, and optimize and synthesize the gene sequence according to the codon of E. coli (lines 114-116).

L114-116. The authors should describe the details of positive control sera for GETV, JEV, PEDV, PRV, CSFV and PRRSV (i.e. experimentally-infected pigs or naturally-infected pigs), with the concrete evidence of infection with these viruses (i.e. RT-PCR, virus-isolation, etc.).

Authors’ response:Thank you for your suggestion. We found the referee’s comments most helpful and have revised the manuscript. Modifications are as follows. JEV (Japanese encephalitis virus), PRRSV (Porcine Reproductive and Respiratory Syndrome), CSFV (Classical swine fever virus), and PRV (Pseudorabies virus), PEDV (porcine epidemic diarrhea virus) positive serum was from experimentally-infected pigs and provided by the China Animal Health and Epidemiology Center (Shanghai Branch).20 GETV positive or negative sera collected from domestic pigs were screened and validated by the virus-neutralization (VN) test [13] or IFA [31] for optimization , specificity, sensitivity of the ELISA protocol (Supplementary table S1). We have corrected this comment in our revised manuscript (lines 97-104).

L110, L249-250. Lack of information again, for the 100 field sera used for the ROC analysis (geographical location, single farm or multiple farms, age, clinical status, etc.). Are they distinct from the 133 sera tested in Tables 3 and 4? The authors should describe the details in Materials and Methods section. Moreover, the seropositivity/negativity was determined by the Western blot analysis, whose diagnostic accuracy has also not been validated. The authors should add the result of VN test for these sera, and evaluate the sensitivity, specificity and coincidence with the rE2-ELISA.

Authors’ response:Thank you for your suggestion. We found the referee’s comments most helpful and have revised the manuscript. Modifications are as follows. First, to determine the cutoff value for the ELISA, 100 sera which were randomly selected from 133 sera were tested simultaneously (lines104-105). Second, GETV positive or negative sera collected from domestic pigs were screened and validated by the virus-neutralization (VN) test [13] or IFA [31] for optimization , specificity, sensitivity of the ELISA protocol (Supplementary table S1). We have corrected this comment in our revised manuscript (lines 97-104).

According to our previous report, immunofluorescent assay (IFA)can also be used for the detection of GETV antibodies(Hameed M,et al. 2021).According to suggestions,we selected 20 serum samples, including the serum used to verify the sensitivity and specificity of the rE2-ELISA and partial serum from the coincidence of the rE2-ELISA, and tested them by VN test and IFA test in order to compare the consistency of the two methods,The results show that the two methods are consistent in detecting GETV antibody(Supplementary table S1). We have corrected this comment in our revised manuscript (lines 101-104).

L170-171. The authors should provide a reference for the principle that “the P/N ratio should be over 2.1”.

Authors’ response:Thank you for your suggestion. We found the referee’s comments most helpful and have revised the manuscript. Modifications are as follows. We added a reference to illustrate this problem. All these results were based on the principle that the P/N ratio should be > 2.1[28] (lines 215-216).

Reference:

  1. Shan, Y.; Liu, Y.; Liu, Z.; Li, G.; Chen, C.; Luo, H.; Chen, Y.; Guo, N.; Shi, X.; Zhang, X.; Fang, W.; Li, X. Development and application of an indirect enzyme-linked immunosorbent assay using recombinant S1 for serological testing of porcine epidemic diarrhea virus. Can J Microbiol. 2019, 65, 343-352.

L201-203. This experiment (Fig. 3C) does not make any sense. I suggest to remove this.

Authors’ response: Thank you for your suggestion. We found the referee’s comments most helpful and have revised the manuscript. Modifications are as follows. We had deleted Fig. 3C and changed in Fig. 2C (lines 260).

L211-214 (Table 3). Very confusing. If it is a part of validation of rE2-ELISA, the authors should indicate the sensitivity and specificity as well. Or, if it is a part of serosurveillance, we can get no information from the coincidence of rE2-ELISA and IFA. Overall, the use of two sets of field sera for validation of ELISA (100 sera in Fig. 2A and 133 sera in Table 3) is irrational.

Authors’ response:Thank you for your suggestion. We found the referee’s comments most helpful and have revised the manuscript. First, since we supplemented the results of VN and IFA, we found that the two are consistent, which can explain the use of IFA to identify ELISA. Table 3 is a part of rE2-ELISA, in order to verify its coincidence rate, by comparing with IFA. Specificity and sensitivity assays are 3.4, Fig. 2B and 3.5, Fig. 2C, which are also part of the establishment of the rE2-ELISA. Second, the reason why 133 sera were selected in Table 3, while 100 sera were used in Fig. 2A is that Fig. 2A is to determine the critical value of rE2-ELISA, so 100 sera were selected, if all sera were selected, the workload is too large, and if the number of selected serum is too small, the problem cannot be explained; and Table 3 is to verify the coincidence rate, the more serum, the better the accuracy of the test, so all serum is used.

L234. “GETV is mainly distributed in southern cities of China” The authors should provide references.

Authors’ response:Thank you for your suggestion. We found the referee’s comments most helpful and have revised the manuscript. Modifications are as follows. We had added references. GETV is mainly distributed in southern cities of China [12,13,27] (lines 295).

L247-248. “We established an ELISA based on the rE2 protein of GETV”. Overstatement.

Authors’ response:Thank you for your suggestion. We found the referee’s comments most helpful and have revised the manuscript. Modifications are as follows. We changed " established " to "improved". In the present study, we improved an ELISA based on the rE2 protein of GETV (lines 309-310).

L255. “newly developed GETV-E2 ELISA”. This ELISA was developed by the other groups and is not the original achievement of this paper. The authors should state that they re-confirmed the usefulness of the rE2-ELISA, providing appropriate references.

Authors’ response:Thank you for your suggestion. We found the referee’s comments most helpful and have revised the manuscript. Modifications are as follows. We changed into “These results suggested that we re-confirmed the usefulness of the rE2-ELISA, providing appropriate references for both the diagnosis of GETV infection and seroepidemiological surveys (lines 316-318)”.

Reviewer 2

In this study the authors developed in house ELISA assay for detection of antibody against Getah virus in pigs. The authors cloned 46 KDa protein of E2 protein and they used different coating conc and defined that 24.5 μg/mL is the best coated conc.

After that the authors analyzed serum samples related to this virus or different viruses to detetmine the assay senstivity and specifcity.

There are many missing informations in this study.

1- During optimization of ELISA, the authors should have positive control to in eact step. Unfortuntaley, there is not positive controls mentioned in this study.

Authors’ response:Thank you for your suggestion. We found the referee’s comments most helpful and have revised the manuscript. Modifications are as follows: 20 GETV positive or negative sera collected from domestic pigs were screened and validated by the virus-neutralization (VN) test [13] or IFA [31] for optimization , specificity, sensitivity of the ELISA protocol (Supplementary table S1). We have corrected this comment in our revised manuscript (lines 97-104).

2- It is clear and not mentioned how the authors detemine the working conc for detection antibody, incubation temperature for coating/detection antibodies, these finding mentioned in the abstract.

Authors’ response:Thank you for your suggestion. We found the referee’s comments most helpful and have revised the manuscript. Modifications are as follows. We changed into “By using chessboard titration and comparing the p/n values, we found that the optimal concentration of coated antigen was found to be 24.5 μg/mL, and the optimal dilution of serum specimens was 1:100. The best working dilution of the horseradish peroxidase (HRP)-conjugated goat anti-pig immunoglobulin (IgG) was 1:5,000. The optimal coating conditions were 12 h at 4°C. The optimal incubation conditions for serum specimens, blocking, and reaction with the secondary antibody were all 1 h at 37°C (lines 35-41).”

3- It is not mentioned how many samples used to develop data in table 1 which is the main key element in this study.

Authors’ response:Thank you for your suggestion. We found the referee’s comments most helpful and have revised the manuscript. JEV (Japanese encephalitis virus), PRRSV (Porcine Reproductive and Respiratory Syndrome), CSFV (Classical swine fever virus), and PRV (Pseudorabies virus), PEDV (porcine epidemic diarrhea virus) positive serum was from experimentally-infected pigs and provided by the China Animal Health and Epidemiology Center (Shanghai Branch). 20 GETV positive or negative sera collected from domestic pigs were screened and validated by the virus-neutralization (VN) test [13] or IFA [31] for optimization , specificity, sensitivity of the ELISA protocol (Supplementary table S1). We have corrected this comment in our revised manuscript (lines 97-104).

4- Why the authors compare data of ELISA with data of IFA. PCR should be the gold standard, the authors need to evlaute the study in PCR positive samples.

Authors’ response:Thank you for your suggestion. We found the referee’s comments most helpful. The ELISA method established in this paper is mainly used to detect GETV antibodies. According to our previous report,PCR is only suitable for the detection of viral nucleic acid, however, there is a transient viremia in the early stage of arbovirus infection (Zhang Y,et al. 2022).thus PCR results are sometimes inconsistent with ELISA results.Indeed,the virus-neutralization (VN) test is the gold standard for GETV antibody detection. According to our previous report, immunofluorescent assay (IFA)can also be used for the detection of GETV antibodies(Hameed M,et al. 2021).According to suggestions,we selected 20 serum samples, including the serum used to verify the sensitivity and specificity of the rE2-ELISA and partial serum from the coincidence of the rE2-ELISA, and tested them by VN test and IFA test in order to compare the consistency of the two methods,The results show that the two methods are consistent in detecting GETV antibody(Supplementary table S1). We have corrected this comment in our revised manuscript (lines 101-104).

Reference:

Zhang Y, Li Y, Guan Z, et al. Rapid Differential Detection of Japanese Encephalitis Virus and Getah Virus in Pigs or Mosquitos by a Duplex TaqMan Real-Time RT-PCR Assay. Front Vet Sci. 2022 Apr 7;9:839443.

5- Figure 3, how did the authors detemine that The cutoff value was determined to be 0.344. What equation used? please add details.

Authors’ response:Thank you for your suggestion. We found the referee’s comments most helpful. Here is my explanation. Using analyze -roc curve in SPSS 22.0, the ROC analysis showed that the area under the curve (AUC) for the ELISA was 0.964 (95% confidence interval [CI], 0.940−0.989; Fig. 2A), and the sensitivity and specificity were 97.1% and 93.3%, respectively. The cutoff value was determined to be 0.344[29,30] (lines 224-228).

Reviewer 2 Report

In this study the authors developed in house ELISA assay for detection of antibody against Getah virus in pigs. The authors cloned 46 KDa protein of E2 protein and they used different coating conc and defined that 24.5 μg/mL is the best coated conc.

Afterthat the authors analyzed serum samples related to this virus or different viruses to detetmine the assay senstivity and specifcity.

There are many missing informations in this study.

1- During optimization of ELISA, the authors should have positive control to in eact step. Unfortuntaley, there is not positive controls mentioned in this study.

2- It is clear and not mentioned how the authors detemine the working conc for detection antibody, incubation temperature for coating/detection antibodies, these finding mentioned in the abstract.

3- It is not mentioned how many samples used to develop data in table 1 which is the main key element in this study.

4- Why the authors compare data of ELISA with data of IFA. PCR should be the gold standard, the authors need to evlaute the study in PCR positive samples

5- Figure 3, how did the authors detemine that The cutoff value was determined to be 0.344. What equation used? please add details

Author Response

(The authors gave the same response as above.)

Round 2

Reviewer 1 Report

In the comments for the original submission, I pointed the lack of information about GETV positive/negative sera used for validating the assay: whether they were from experimentally-infected pigs or naturally-infected pigs; whether they were taken from one pig or pooled sera from multiple pigs; VN titers, etc. Particularly, if the control sera were collected from naturally-infected pigs, the concrete evidence of GETV infection in these animals should be described.

In the current submission, the authors added information of “20 GETV positive or negative sera” with the result of IFA, VN and the ELISA (Supplementary material). However, it is still unclear which specific serum or pooled sera were used for each experiment (Fig. 1C, Fig. 2B and 2C, Table 1). For example, there was no range information for the OD values in Table 1, which may suggest only one positive and one negative serum might be used. Fig. 2B was shown with error bars and multiple pig sera might be used, and Fig. 2C should come from one specific positive serum, but there was no description at all for which specific ones among 20 sera were used.

A more fundamental problem is that, these 20 sera were field samples with no information of infection status (any test results aside from serology that can directly prove GETV infection in these pigs, i.e., RT-PCR or virus-isolation, etc.), and therefore, it is inappropriate to use them as the control sera. As this paper focused on improving and optimizing the experimental settings of the ELISA, the infection status of pigs from which the control sera were taken is the most critical point. Due to the uncertainty in this point, unfortunately, I would not recommend this paper for publication.

Reviewer 2 Report

I do not have any further concerns